# CXray-EffDet: Chest Disease Detection and Classification from X-ray Images Using the EfficientDet Model

**DOI:** 10.3390/diagnostics13020248

**Published:** 2023-01-09

**Authors:** Marriam Nawaz, Tahira Nazir, Jamel Baili, Muhammad Attique Khan, Ye Jin Kim, Jae-Hyuk Cha

**Affiliations:** 1Department of Computer Science, University of Engineering and Technology, Taxila 47050, Pakistan; 2Department of Software Engineering, University of Engineering and Technology, Taxila 47050, Pakistan; 3Faculty of Computing, Department of Computer Science, Riphah International University Gulberg Green Campus, Islamabad 04403, Pakistan; 4College of Computer Science, King Khalid University, Abha 61413, Saudi Arabia; 5Higher Institute of Applied Science and Technology of Sousse (ISSATS), Cité Taffala (Ibn Khaldoun) 4003 Sousse, University of Souse, Sousse 4000, Tunisia; 6Department of Computer Science, HITEC University, Taxila 47080, Pakistan; 7Department of Computer Science, Hanyang University, Seoul 04763, Republic of Korea

**Keywords:** deep learning, EfficientDet, X-ray, chest diseases, classification, localization

## Abstract

The competence of machine learning approaches to carry out clinical expertise tasks has recently gained a lot of attention, particularly in the field of medical-imaging examination. Among the most frequently used clinical-imaging modalities in the healthcare profession is chest radiography, which calls for prompt reporting of the existence of potential anomalies and illness diagnostics in images. Automated frameworks for the recognition of chest abnormalities employing X-rays are being introduced in health departments. However, the reliable detection and classification of particular illnesses in chest X-ray samples is still a complicated issue because of the complex structure of radiographs, e.g., the large exposure dynamic range. Moreover, the incidence of various image artifacts and extensive inter- and intra-category resemblances further increases the difficulty of chest disease recognition procedures. The aim of this study was to resolve these existing problems. We propose a deep learning (DL) approach to the detection of chest abnormalities with the X-ray modality using the EfficientDet (CXray-EffDet) model. More clearly, we employed the EfficientNet-B0-based EfficientDet-D0 model to compute a reliable set of sample features and accomplish the detection and classification task by categorizing eight categories of chest abnormalities using X-ray images. The effective feature computation power of the CXray-EffDet model enhances the power of chest abnormality recognition due to its high recall rate, and it presents a lightweight and computationally robust approach. A large test of the model employing a standard database from the National Institutes of Health (NIH) was conducted to demonstrate the chest disease localization and categorization performance of the CXray-EffDet model. We attained an AUC score of 0.9080, along with an IOU of 0.834, which clearly determines the competency of the introduced model.

## 1. Introduction

Easier accessibility to multimodel content, i.e., digital samples and audiovisual data, has boosted the development of jobs in the area of computer vision (CV). Some famous areas of CV include object identification and tracking [1] and computer-aided analysis of several medical imaging techniques [2]. Professionals are being aided by the use of CV techniques in the analysis of medical images to complete tasks rapidly and correctly. Chest X-ray (CXR) examinations are one of these applications [3]. In order to diagnose various respiratory anomalies, including pneumonia, COVID-19, bronchiectasis, lung lesions, etc., chest X-ray imaging is the modality that is most frequently used worldwide [4]. Important clinical inspections are performed daily as a result of the chest X-ray modality’s simpler and more practical approach [5]. However, the accessibility of the subject matter to experts is crucial for the manual review of these samples. Additionally, manual inspection of chest CXR research is a time-consuming, challenging task with a significant likelihood of inaccurate findings. Contrarily, a computerized recognition model can expedite the procedure while also improving the system’s effectiveness.

Around 65 million individuals worldwide are afflicted by one or more types of chest diseases, and 3 million people pass away each year as a result of such illnesses. Thus, early detection of these abnormalities can spare individuals’ lives and prevent them from invasive surgical treatments [6]. Consequently, scientists have concentrated their efforts to propose trustworthy computerized alternatives to address the issues with manual CXR inspection. Hence, artists have concentrated their efforts to propose trustworthy automated alternatives to address the issues with the manual CXR examination process. Initially, various CXR anomalies were classified using hand-crafted pattern calculation techniques. These procedures are straightforward and effective with tiny amounts of data. Hand-crafted key feature-extraction methods, in comparison, take a long time and require a lot of subject knowledge to generate correct answers. Additionally, there will always be a trade-off between categorization accuracy and computational cost for such algorithms. These approaches’ ability to recognize objects can be improved by using large keypoints, but at the expense of extra processing costs. Though using smaller keypoint sets makes hand-coded techniques more effective, it leaves out important aspects of visual modalities, which lowers the precision of classification outcomes. These factors prevent these approaches from being effective for CXR evaluation [7]. 

The progress of AI-based solutions in the automated identification of medical problems is astounding at the moment. When used in medicine, AI aids with patient management, diagnosis, and treatment. This relieves physicians’ burden and lends a helping hand to doctors. Such frameworks also provide assistance to the administration department of a healthcare unit by providing automated management solutions [8]. The scientific world has progressively become more engaged in using deep learning (DL) methods for digital image processing, along with the CXR test. To accomplish the segmentation and categorization task involved in several medical disease-related issues, a variety of well-studied DL models are used, such as convolution neural networks (CNN) [9] and recurrent neural networks (RNNs) [10]. As a result, deep learning is becoming an extremely potent solution in the health sector, since the majority of tasks are related to the classification and segmentation of diseases. Due to the empowerment of DL techniques, these systems are ideally suited for analyzing medical images, since they can compute a more nominative set of image characteristics without the need for subject-matter specialists. The way in which people’s brains view and remember different objects serves as an inspiration for Cnn architectures. A few such frameworks are VGG [11], GoogleNet [12], ResNet [13], XceptionNet [14], DenseNet [15], and EfficientNet [16], etc., which are being thoroughly explored in the field of medical image analysis. Such techniques can deliver consistent results with little computation. The key rationale behind employing DL-based algorithms for the computer-aided diagnosis of medical sample investigations is that they have the ability to compute the essential data of the input sequence and can handle challenging sample distortions, i.e., luminance and chrominance changes, clutter, blurring, and size changes, among others.

Extensive research has been carried out in the domain of the recognition of chest abnormalities. In this section, a detailed analysis of existing work is accomplished to recognize the various abnormalities of the chest. One method was discussed in [3], in which the DL methods Vgg-16 and XceptionNet were employed for locating pneumonia-affected areas of images. In the first step, data augmentation was performed with various operations such as zoom, angle rotation, and image flip to enhance the diversity of the database. In the next phase, the DL approaches were applied to compute a set of deep keypoints. The work gained the highest results for the XceptionNet model; however, categorization performance requires more enhancement. Bhandary et al. [17] also introduced a model to recognize pneumonia-affected areas from lung samples by employing DL approaches. Initially, a model called custom AlexNet was introduced for pneumonia detection. In the second step, an ensemble approach was used to join pattern-based keypoints computed via the application of the Haralick and Hu algorithm [18] with the deep keypoints calculated with the first model. The computed keypoint set was utilized to train a support vector machine (SVM) and softmax classifier. The work in [17] acquired categorization results of 97.27% utilizing CT samples over the LIDC-IDRI repository. In [19], the researchers used various image dimensions and transfer learning to assess the effectiveness of various pre-trained deep networks, including GoogleNet, InceptionNet, and ResNet. Additionally, the properties that these models learned were analyzed using network visualization. According to the findings, shallower nets, such as GoogleNet, perform better than deep networks at differentiating between healthy and diseased chest x-rays. A DL-based approach called CheXNet was developed by Rajpurkar et al. [20] to diagnose various disorders in the chest. Using batch normalization and dense connection, the network had 121 layers. The ChestX-ray14 database was used to reinforce the ChexNet classifier, which was built earlier with ImageNet samples. This method received an F1 measure of 0.435 and an AUC of 0.801. A DL approach for coronavirus disease identification over a variety of other chest ailments using chest X-rays was put forth by the researchers in [21]. To address the concerns with skewed class samples, the authors used a GAN-based method to produce synthetic images. They used a variety of scenarios to analyze the model behavior, including data augmentation, transfer learning, and unbalanced category samples. The outcomes demonstrated that, with balanced data, the ResNet model produced a high performance of 87%. Ho et al. [22] developed a two-phase method to accurately identify 14 distinct disorders from X-ray samples of the chest. Utilizing activation weights taken from the final convolutional layer of the trained DenseNet121 net, the anomalous area was initially located. Next, categorization was carried out by combining patterns and deep features to perform the fusion of keypoints. Composite characteristics were categorized using a number of supervised learning algorithms, including SVM, KNN, AdaBoost, and others. The experiments revealed that, with an accuracy of 84.62%, the ELM predictor performed well in contrast to other learners. Utilizing chest X-ray scans, the researchers in [23] created a CNN network with 3 convolutional layers to identify 12 distinct disorders. They examined and studied how well unsupervised learning performed compared to backpropagation NN and competitive NN. The outcomes showed that the suggested CNN’s strong feature learning enabled it to achieve a high classification rate and greater generalization capability. However, the number of convergence cycles and calculation time was a little higher.

Another work was discussed in [24] in which the researchers created a multi-scale attention net to improve the efficiency of identifying multiple classes of chest diseases. By employing a multi-stage attention block that combined local attributes obtained at various sizes with global keypoints, the introduced network used Densenet-169 as its core. To address the problem of imbalanced image samples, a unique loss method utilizing perceptual and multi-label balancing was also devised. The AUC of the proposed method was 85 on the CheXpert database and 81.50 on the ChestX-Ray14 repository. A cross-attention-oriented, end-to-end structure was proposed by Ma et al. [25] to overcome the class-imbalanced issue of multi-label X-ray chest disease categorization. For improved keypoint depiction via shared attention, the model included a loss formula containing attention and multi-category balancing loss in addition to the computation of features with Densenet having 121 and 169 layers. On the ChestX-Ray14 samples, the discussed model displayed an improved AUC of 81.70. To increase the precision of multi-class respiratory disease detection using chest radiography, Wang et al. [26] introduced the ChestNet framework. Two sub-networks, a categorization net, and an attention net made up the entire framework. A pre-trained ResNet152 algorithm was utilized to collect uniform keypoints that served as the foundation for the categorization network. Utilizing the keypoints that were retrieved, the attention net was utilized to examine the connection between category labels and anomalous areas. Employing the ChestX-ray14 database, the developed framework performed better for categorization. A strategy to perform multi-label chest illness classification and abnormality localization concurrently was suggested by Ouyang et al. [27]. The paradigm included both an activation- and a gradient-based attention mechanism (AM). The work utilized a hierarchy-based visual AM with three levels: forefront, positive/negative, and abnormal attention. Because there were only a few box annotations for the diseased area that were readily available, the framework was learned by employing a weakly supervised learning approach. The average AUC value for this method on the ChestX-ray14 database was 81.90.

Pan et al. [28] employed the well-known DL networks DenseNet and MobileNet-V2 to differentiate diseased samples of chest X-rays from normal images. The work was focused on classifying 14 types of chest diseases. The approach was trained and tested on two different datasets to test the cross-corpus behavior of the model. The work discussed in [28] attained the best results with the MobileNet-V2 model. Another technique was presented in [29] focusing on classifying both coronavirus and chest-related diseases from the X-ray image modality via the employment of the DL approach. In the first phase, samples were categorized as healthy, coronavirus-affected, and other, which were later distributed into 14 classes of chest abnormalities. The work utilized various deep nets, and ResNet-50 attained the best results in the recognition of COVID-19 and other chest disease-affected samples. A technique for the identification of both bacterial and viral influenza using chest images was developed by Alqudah et al. [30]. First, a customized CNN technique was tailored to learn characteristics unique to the pneumonia illness after being pre-trained on other medical samples. Next, categorization was carried out with various predictors, including the KNN, SVM, and softmax algorithm. The findings demonstrated that the SVM significantly improved results compared to the other classifiers, but the effectiveness was assessed using a small sample set. An end-to-end training method for performing multi-class respiratory illness categorization was described in [31]. In order to eliminate irrelevant information from samples, a preprocessing step was initially applied using trim and resize procedures. In order to construct a nominative feature set and subsequently classify the images into the appropriate categories, the pre-trained EfficientNetv2 network was applied. This strategy yields better outcomes for three-class classification, but the algorithm has over-fitting issues and performs worse as the classes increase. The effectiveness of various ResNet-based methods for the problem of labeling multiple chest X-ray data was examined by Baltruschat et al. [32]. For better classification, the scientists expanded the design and added non-image information, including the patient’s age, gender, and sample capture category, to the net. According to the findings, the ResNet model with 38 layers integrating metadata outperformed the rest of the networks, with an AUC of 72.70. A DL approach performing multi-category identification of chest diseases from X-ray and CT samples was proposed in [33]. Four distinct customized structures built on VGG-19, V2-based ResNet-152, and Gated Recurrent Unit (GRU) were evaluated by the researchers. The findings demonstrated that the customized VGG-19 model beat the other algorithms by achieving an accuracy rate of 98.05% on the X-ray and CT data samples; however, the method had problems with model over-fitting. Ge et al. [34] developed a multi-class technique for diagnosing diseases from chest X-rays making use of labeling dependencies for illnesses and states of health. The network was made up of two separate sub-CNNs that learned with multiple loss method pairings, including correlation, multi-label softmax, and binary cross-entropy losses. Bilinear pooling was additionally implemented by the researchers to calculate useful keypoints for effective categorization. With ResNet as the backbone model, this technique [34] displayed an AUC of 83.98; however, it had a high complexity of computation. Albahli et al. [35] proposed an approach for classifying chest diseases using the X-ray image modality. Initially, samples were preprocessed to improve the visual appearance. After this, a histogram of the gradient (HOG) descriptor was used with numerous DL frameworks such as ResNet101, DenseNet201, etc. for features computation. Meanwhile, the SVM technique was used for classification. The approach acquired the best results with the ResNet101 model, but at the expense of increased computational cost.

The research mentioned above has produced impressive results. However, these approaches can only be used to classify a small number of chest-related disorders, because they are not generalizable. Table 1 provides a summary of the methods for identifying chest illnesses from the literature. As can be observed, there is still room for advancement in regard to categorization accuracy, computing overhead, and generalization power for multi-label chest diseases.

Although current methodologies have produced impressive CXR recognition accuracy, there is a need for progress in terms of precision and computational burden. As a result, a deeper analysis of the classical machine learning (ML) and DL methodologies are needed to improve the effectiveness of classifying chest-related disorders from the X-ray medical image modality. The main issue with ML approaches for categorizing CXR abnormalities is their decreased efficacy with longer computation times [36]. Comparing DL techniques to human brain intelligence, the capacity to resolve challenging real-world problems is astounding. Even though the DL method fixes issues with ML techniques, it also made models more sophisticated. Consequently, a more reliable method of classifying CXR-related diseases is required.

Due to the many commonalities between various chest disorders, it might be challenging to classify various CXR disorders accurately and on time. Additionally, the input samples’ prevalence of distortion, fading, illumination variance, and brightness shifts makes the categorization process even more challenging. Therefore, we developed a new framework called chest abnormalities’ detection from the X-ray modality using the EfficientDet (CXray-EffDet) model to locate and categorize the eight different categories of chest anomalies in order to address the shortcomings of previous approaches. Descriptively, we utilized the EfficientNet-B0-based EfficientDet-D0 model to compute a reliable set of sample features and accomplish the classification task by categorizing the eight categories of chest illness from the X-ray images. The experimental findings demonstrate that even with the existence of diverse image distortions, our method is competent at differentiating between the different kinds of chest disorders. Below are the primary contributions of the proposed work:A model named the CXray-EffDet is proposed to recognize chest diseases from X-ray images.The introduced framework is proficient in correctly identifying and categorizing the eight kinds of chest abnormalities from the X-ray images due to the robustness of the CXray-EffDet approach.The proposed model boosted the categorization performance due to its enhanced recognition ability to tackle complex sample transformation changes.The CXray-EffDet approach presents a computationally effective approach to classifying chest disorders from X-ray samples due to its one-stage object identifier.The proposed work is capable of both detecting the locations of diseased regions and their associated class.To demonstrate the accuracy of our method, we give a detailed assessment of the new methods for the categorization of CXR diseases and perform extensive experiments on a challenging database called NIH Chest X-ray.

The rest of the article follows the above outline. An explanation of the presented work and employed dataset is given in Section 2. Section 3 comprises the description of the employed performance metrics and experiments used to assess the classification results of the presented work. Section 4 comprises the discussion, and the conclusion is elaborated on in Section 5.

## 2. Materials and Methods

The proposed work consists of two major phases, which are divided into: (i) the preparation of image samples as per model requirements and (ii) the identification and classification of numerous chest X-ray diseases. A visual representation of the proposed approach is given in Figure 1. In the first phase, the samples are annotated by drawing a rectangular box around the diseased region in the X-ray samples to precisely determine the region of interest. Then, in the next step, the annotated images are utilized to train a DL approach called chest abnormalities’ detection from the X-ray modality using the EfficientDet (CXray-EffDet) model. More clearly, we have employed the EfficientNet-B0-based EfficientDet-D0 framework to compute a distinct set of sample keypoints and accomplish the classification task by categorizing the 8 classes of chest disorders from the X-ray samples. The CXray-EffDet framework performs three steps to execute the localization and categorization task. Initially, the features extractor of the CXray-EffDet approach named the EfficientNet-B0 accepts the image as input and computes a deep set of sample features. Then, the BiFPN unit executes top-down and bottom-up feature extraction, blending numerous times to compute the final set of keypoints at Levels 3 to 7 in EfficientNet. In the last phase, the identified diseased area along with the predicted class label is shown, and model performance is calculated by utilizing the evaluation metrics used in the domain of computer-aided medical image processing. Thorough details of the introduced model are given in Algorithm 1.
**Algorithm 1:** Infected Region DetectionINPUT:TrS, BBxOUTPUT:Identified Area, EfDet, Class*TrS—*total images used for model training.*BBx—*coordinates of the rectangular box showing the diseased portion.Identified Area—diseased portion in the output.*EfDet –*EfficientDet model with the EfficientNet-B0 base network.Class portion—Label indicating the category of each identified area.Size_of_Sample ← [x y]// Computing region of interest      *α*← AnchorsComputation(*TrS*, BBx)// EfDet-Approach     *EfDet* ← EfficientDet_B0Base (Size_of_Sample, α)      [*P_tr_ P_te_*] ← Distribution of the employed repository in the *train* and *test* parts// Training phaseFor each image *i* in → *P_tr_*Compute *EffNet(B0)-*features→*df*Compute *EffNet(B0)-*features→*df*Accomplish keypoints Fusion (*df)* →*Ff*EndTrain *EffDet* using *Ff*, and measure execution time *t_EffDet**η_EffDet*← LocateAffectedRegion(*Ff*)*Ap_EffDet* ← Evaluate_AP (*EffNet(B0)*, *η_ EffDet)*For each sample *j in* → *P_te_*a) Extract key features through trained model Ap_EffDet →β*I* b) [Bbox, ConfidenceScore, category] ←Estimate (β*I*) c) show the image with *Bbox*, score, and *class label* d) *compute*AccuracymAPprecisionrecalltime*End For*

### 2.1. Matetial

To perform the training and evaluation of the CXray-EffDet model, a standard and publicly available database, the NIH Chest X-ray, is used in the introduced work [37]. The dataset contains 112,120 images of 30,805 subjects. There are a total of 14 diverse categories of chest abnormalities in the NIH repository, as depicted in Figure 2. However, expert-provided annotations are available for only 8 classes of diseases, which are as follows: atelectasis, cardiomegaly, effusion, infiltration, mass, nodule, pneumonia, and pneumothorax, respectively. Therefore, as the CXray-EffDet model is concerned with both localizing and classifying chest diseases, we have used the mentioned 8 chest abnormalities in the presented work. This dataset contains a total of 984 annotated images that are used for model training. The basic reason for selecting the NIH CXR dataset in this work is that the samples of this repository are complicated and diverse and contain several transformations, such as variations in light, color, and size of diseased region, as well as the incidence of noise, blurring, etc. 

### 2.2. Annotations

It is crucial to explicitly illustrate the location of the diseased areas in the suspected samples for an effective and appropriate method of training. We used the LabelImg [39] tool to produce annotations of the diseased image regions and precisely define the ROIs to complete this operation. The created annotations are saved in a file that contains two different sorts of data: the coordinates of the Bbox showing the affected area from the X-ray chest images and the class that is assigned to each spot that is found. The model learning file is then created from the XML file and used to build the network.

### 2.3. CXray-EffDet

In the proposed method, we present a DL framework called the CXray-EffDet network that employs the EfficientDet model for chest disorders’ classification using the X-ray images. To appropriately identify the chest diseases from the X-ray images, an effective feature extractor is required. However, due to the reasons listed, acquiring a more relevant group of image attributes is a difficult task. The estimation of a larger-sized feature set could lead to the model over-fitting issue, whereas a smaller-sized feature map could prevent the model from learning some crucial sample components, such as color and shape changes, which blur the distinctions between normal and affected sample regions of an image. It is crucial to adopt an automated keypoint estimation methodology instead of a hand-crafted feature-calculation strategy in order to obtain a more reliable set of sample keypoints. Due to wide differences in the mass, shape, luminance, and location of chest X-ray diseases, the architectures using hand-coded feature estimation are ineffective in properly finding and categorizing diseased regions in their respective classes. We used EfficientDet [1,40], a DL-based method with the ability to automatically calculate the resilient key features from the images under analysis, to address the above-elaborated problems. EfficientDet’s convolution filters analyze the structural details of the input sample to regulate the characteristics of the diseased regions. In order to locate and identify different medical disorders, scientists have offered a number of object-detecting techniques. These methods are broadly categorized into two types: one-stage (RetinaNet [41], YOLO [42], CornerNet [2], SSD [15], CenterNet [43]) and two-stage (RCNN [44], Mask-RCNN [45], Fast-RCNN [11], Faster-RCNN [21]) approaches. 

The reason for choosing EffieicntDet over other one-stage detection techniques is that these methodologies indicate a minimal time to complete the classification task, which compromises the prediction performance. Though two-stage detection models are more accurate at detecting medical abnormalities, this improvement comes at the expense of high computation complexity, because these methods require two phases to find and categorize ROIs, making them inappropriate for use in real-life situations. Therefore, it is necessary to propose a strategy that would result in the identification and classification of chest X-ray diseases in a robust and efficient manner. We used the EfficientDet technique, introduced by the Google Brain team, to resolve the aforementioned problems. The EfficientDet framework is a robust and reliable object-recognition technique that builds on the multi-directed feature unification structure of FPN and takes inspiration from the scaling strategy of the EfficientNet structure. The proposed CXray-EffDet model contains 3 major components, where the first unit is the EfficientNet model employed for features extraction. In the presented approach, the EfficientNet-B0 is nominated as the base network of the CXray-EffDet model. The second unit is termed the BiFPN, which accomplishes both topmost and lower-right keypoint blending numerous times for the calculation of the final feature set. Finally, the last unit of the proposed work is concerned with recognizing and categorizing the identified portion into eight different types of chest diseases. A thorough explanation of training parameters utilized for model training is elaborated in Table 2.

#### 2.3.1. EfficientNet-B0

For the purpose of obtaining the deep keypoints vector from the input samples, we chose EfficientNet -B0 as our backbone model. The EfficientNet approach consistently adjusts all dimensions with a predetermined set of scalability factors, as opposed to conventional techniques that change model sizes, i.e., breadth, depth, and resolution, arbitrarily. The EfficientNet -B0 is able to compute a more effective set of imaging keypoints with a limited range of parameters, which also reduces calculation time and increases detection performance. Figure 3 shows the organization of the EfficientNet-B0 model. The EfficientNet architecture is able to appropriately portray sophisticated image transformation, which allows it to cope with the challenge of the lack of ROI positional information more effectively. The EfficientNet architecture also permits the reuse of calculated features, making it more appropriate for chest illness diagnosis and accelerating the training process.

#### 2.3.2. BiFPN

In the effective identification and recognition of various chest X-ray diseases, characteristics such as location, background, intensity changes, and mass of the diseased portion must be accounted for. As such, the employment of a multi-scale features extraction strategy can be effective for the accurate recognition of numerous categories of chest disorders from the X-ray samples. Frameworks historically have typically used top-down FPNs to combine multi-stage features. The single-directed FPN, however, does not necessarily contribute to different scales proportionally in the resulting features, which might lead to the failure to acquire some crucial image behaviors in the chest abnormalities’ detection processes.

In order to more effectively address the issue of equitable participation in FPN, the notion of BiFPN is included in the method that is currently presented. The BiFPN unit uses regular and consistent links to enable information to pass in both the topmost and lower-right sides. Additionally, the BiFPN component employs trainable weights to select relevant keypoints with important contributions to the final architecture. Hence, features from the P-(3 to 7) layers of the EfficientNet-B0 are designated as multi-stage keypoints, which are taken as input by the BiFPN component. The breadth of the BiFPN unit increases exponentially with an increase in depth enhanced by satisfying Equation (1):(1)$${ѿ}_{b}=64.\;(1.35^{\emptyset }),\;d_{b}=3+\emptyset$$

In Equation (1), ${ѿ}_{b}$ and $d_{b}$ indicate the breadth and depth of the BiFPN unit, and ∅ is the composite component used to monitor the scaling sizes with a value of 0 in the presented work.

#### 2.3.3. Box/Class Estimation Module

The collective multi-stage features computed by the BiFPN unit are propagated as input to the box/class estimation unit to draw a rectangular box around the detected area along with the respective category. The size of this unit is similar to the BiFPN module, and depth is calculated as:(2)$$d_{Recbox}=3+\left[\emptyset /3\right]$$

#### 2.3.4. Detection Process

The CXray-EffDet framework is a more effective method in comparison to other approaches and does not employ hand-coded feature computation methods, i.e., selective search and proposal generation. As such, the input sample is passed to the trained model, on which it directly locates the location of the diseased region by drawing a bounding box around it and determines the associated class label along with the confidence score. 

## 3. Experiment and Results

Here, we discuss the performance measures that were utilized to numerically estimate the obtained results. Furthermore, we carried out an extensive experimental evaluation to assess the proposed work in several ways to explain the efficacy of the CXray-EffDet framework for locating and recognizing the eight kinds of chest abnormalities from the X-ray images. The proposed work is implemented in a GPU-based system with the Nvidia GTX1070 card in the Python language. In the proposed work, the CXray-EffDet model employs the learned weights gained over the MS-COCO database, and transfer learning is accomplished on the images from the employed repository to train it for recognizing numerous chest abnormalities.

### 3.1. Evaluation Measures

To validate the CXray-EffDet model in localizing and classifying the eight kinds of chest disorders, we selected numerous standard measures that are extensively utilized by researchers working in the domain of object identification and categorization. More clearly, the Intersection-over-Union (IOU), mean average precision (mAP), precision, accuracy, and recall were employed for analyzing the performance of the presented approach. The numerical details to compute the accuracy are elaborated in Equation (3): (3)$$Accuracy=\frac{TP+TN}{TP+FP+TN+FN}$$

The *mAP* is defined in Equation (4), in which *AP* defines the average precision over all categories, and *t* represents a test image. Moreover, *T* shows total evaluation images.
(4)$$mAP:=\sum_{i=1}^{T}AP(t_{i})/T$$

Figure 4, Figure 5 and Figure 6 explain the IOU, precision, and recall measures, respectively.

### 3.2. Localization Performance

A robust chest disease detection approach must be competent to reliably identify the affected region and specify the correct class. Therefore, we performed an analysis to validate the localization power of the CXray-EffDet model. For this reason, we have taken the images from the test set of the NIH Chest X-ray repository to evaluate the localization ability of the presented work and attained results in terms of the pictorial demonstration shown in Figure 7. The images exhibited in Figure 7 clearly portray that the CXray-EffDet model is robust in locating the affected regions and can distinguish numerous chest abnormalities competently. We report the results in terms of the numeric form by computing the mAP score, as it is the standard measure used to compute the localization power of an object-detection approach. More descriptively, we obtained an average mAP score of 0.926. Both the graphical and numeric scores clearly show that our work is proficient at locating the various categories of chest diseases with a high recall rate.

### 3.3. Class-Wise Evaluation

In this part of the manuscript, a detailed evaluation of the category-wise results is carried out to explain the recognition ability of the CXray-EffDet model to differentiate the various forms of chest infections employing X-ray samples. To perform this, we took the class-wise samples of the employed data samples and tested them on the trained framework. To assess the performance, we computed the precision and recall scores, along with the accuracy and F1-measure. 

In the first step, we show the category-wise attained precision score of the CXray-EffDet model for all eight categories to validate how effectively the presented model recognizes the various disease classes. The obtained precision scores are given in Figure 8 for all eight classes of chest abnormalities, which clearly show that the CXray-EffDet model is proficient at locating disease samples. More descriptively, the CXray-EffDet model acquired a precision rate of 90%, which exhibits the effectiveness of the proposed approach.

Furthermore, the recall scores for all eight chest diseases are computed to check the recognition capability of the CXray-EffDet framework, and the attained values are given in Figure 9. It can be seen from the performance elaborated in Figure 9 that the CXray-EffDet model is capable of recognizing all categories of illnesses with a high score. In a clearer manner, we gained an average recall value of 92.36%.

Additionally, the F1-score is also reported for the CXray-EffDet model, as this measure helps to provide a better depiction of the performance of a model by showing a comparison of the precision and recall measures. Moreover, we computed the error rates for all categories of chest diseases. The results both in terms of the F1-score and error rate are given in Figure 10. The values depicted in Figure 10 clearly show that we obtained the highest F1-score in the pneumothorax category, with an error rate of 4.67%, and the CXray-EffDet model attained the lowest F1-score and highest error rate for the effusion abnormality, with a value of 89.22%, along with an error rate of 10.78%. Descriptively, the CXray-EffDet model shows an average F1-measure and an error rate of 91.16% and 8.85%, which depicts the robustness of the presented method.

An important aspect of discussing the categorization results of an approach is to show the confusion matrix, as this measure effectively elaborates the categorization results in terms of the attained true-positive rate (TPR), which assists in showing the capability of an approach to locate healthy and diseased classes. Therefore, we computed the confusion matrix for all eight kinds of chest disorders and give the results in Figure 11. The values given in Figure 11 clearly indicate that the proposed model is competent at effectively categorizing all types of chest disorders because of its high recall rate.

Lastly, the category-wise accuracy results were also computed for the CXray-EffDet model, and values are given in Figure 12. The CXray-EffDet model exhibited robust classification performance in terms of the accuracy measure for all eight types of chest abnormalities. Descriptively, we attained an average accuracy value of 94.53%. As per the analysis performed for all eight types of chest diseases in terms of several standard measures discussed in this section, we can say that the CXray-EffDet model is proficient at detecting and classifying chest-related abnormalities. 

### 3.4. Comparative Examination of the Presented Model with Base Approaches

In this section, a category-wise comparative analysis of the CXray-EffDet model is performed with other base techniques to compare the chest disease classification performance with them.

Initially, we nominated the AlexNet [46], GoogleNet [12], VGG16 [47], and ResNet50 [48] models for comparison in terms of the AUC measure, and the acquired results are given in Table 3. The comparison illustrated in Table 3 clearly shows that our work is effective in robustly identifying the various categories of chest diseases as opposed to other approaches. Descriptively, for the atelectasis and cardiomegaly diseases, the comparative methodologies gave AUC values of 65.25 and 72.75 respectively, which were 91.30 and 97.10 for the proposed work. Therefore, for the atelectasis and cardiomegaly abnormalities, we provided performance gains of 26.05%. Similarly, for the effusion and infiltration classes, the comparative models gave AUC scores of 68.50 and 60.25. In the comparison, we attained AUC scores of 94.60 and 79.60 and show performance gains of 26.10% and 19.35%, respectively. For the mass and nodule classes, peer works showed AUC scores of 54.25 and 64.50, which were 92.50 and 85.70 for our work. For the mentioned classes, we have given performance gains of 38.25% and 21.20%, respectively. Moreover, for the pneumonia and pneumothorax diseases, the selected DL approaches attained AUC scores of 57 and 73.50, which were 88.10 and 97.50 for our work, and we attained performance gains of 31.10% and 24%. 

Second, we performed a comparative assessment of the introduced technique in terms of the entire database using numerous standard measures with the base models. The performed analysis is given in Table 4, which clearly shows that our work performs well for all the performance metrics in comparison to the other DL approaches. More descriptively, for the precision metric, the base models showed a value of 71%, which was 90% for our work. As such, we have given a performance gain of 19% for the precision metric. Moreover, for the recall and accuracy measures, the base models attained average scores of 72% and 73%, which were 92.36% and 94.53% for our work and resulted in performance gains of 21% and 22%, respectively. Moreover, for the F1-measure, we showed a value of 91.16%, which was 71% for the base models. Hence, for the F1-measure, the CXray-EffDet model attained a performance gain of 20%.

The performed experimentation both in terms of class-wise results and the entire dataset clearly shows that the CXray-EffDet model is quite proficient at identifying and categorizing the eight classes of chest disorders as compared to other base techniques. The basic cause of the enhanced results of the introduced approach is that it utilizes a shallow framework architecture, which is effective at identifying a large set of sample keypoints. Other base techniques are quite complicated in architecture and unable to tackle image transformations, such as size, orientation, and mass differences of the diseased region, which causes a reduction in the classification performance of these models. In comparison, the CXray-EffDet model can better deal with such changes.

### 3.5. Discussion and Analysis

As the proposed approach is concerned with employing an object-recognition model called the CXray-EffDet framework for the detection and categorization of chest diseases, we performed an analysis to compare its results with other such approaches. To perform this evaluation, both one- and two-stage networks are considered. The primary difference between both object-recognition models is that in two-step approaches, the related class is first established after a large number of area proposals have been produced to pinpoint the position of the diseased component. Meanwhile, for one-stage algorithms, both the location and category of ROIs are computed in a single phase. For two-step techniques, the models of Fast-RCNN [49], Faster-RCNN [50], and Mask-RCNN[51] were nominated, whereas in the case of one-stage methods, the RetinaNet [41] and CenterNet [52] were selected.

To perform a comparative analysis of the categorization performance of all selected approaches, we considered both the detection power and time complexity of all frameworks. To compare the localization results, we chose the mAP measure, as it is the standard measure utilized by object-identification approaches. The conducted analysis comparing both the classification results and time complexities is given in Table 5. The reported results in Table 5 clearly show that the proposed CXray-EffDet model is both effective and competent as compared to all other techniques. More descriptively, for the mAP score, the peer approaches gave an average score of 0.732, which was 0.926 for the proposed work. Therefore, for the mAP metric, we have given a performance gain of 19.40%. Similarly, the proposed approach showed a minimum execution time of 0.20 s in comparison to other object detection models. The basic reason for the improved results of the proposed approach is that the Fast-RCNN utilizes hand-coded models for feature extraction, which are not proficient at handling sample distortions and result in performance degradation.

Techniques such as Faster and Mask-RCNN have resolved the issue of the Fast-RCNN; however, they suffer from high computational complexity because of the two-step object-detection networks. Moreover, the RetinaNet model is not effective at extracting nominative anchors for the acentric features of input images. The CenterNet approach performs well, but is not proficient at tackling unseen cases. Comparatively, the introduced model resolved the problems of the comparated techniques by presenting a more efficient feature descriptor, which deals with the complex pattern of X-ray samples in a more discriminative way and enhances the classification results.

*Comparison with ML-based classifiers*: By comparing the approach’s performance to that of traditional ML-based predictors, we further demonstrated the resilience of our method for diagnosing chest illnesses from X-ray images. Because of this, we chose the SVM and KNN as our two ML classifiers, and the results are given in Table 6. The performance scores listed in Table 6 clearly demonstrate that the given strategy achieved a maximum AUC of 90.80. Meanwhile, the SVM predictor showed the second-best results, with an AUC score of 74.50. The comparable methods displayed an average score of 73.30, which was 0.887 for our technique. Consequently, we delivered a performance gain of 17.50%. The comparative study clearly shows that the CXray-EffDet model is more effective at categorizing the various illnesses of the chest in X-ray medical images.

Lastly, we compared the chest disease identification and categorization results of the CXray-EffDet model with the latest techniques [54,55,56,57,58,59] to evaluate our model’s performance against them. For fair assessment, the average results of these works are compared with the proposed work.

First, the performance was compared in the form of the AUC measure, and an analysis is given in Table 7, which indicates that the CXray-EffDet technique outperformed the other approaches. More clearly, the work in [54] employed a CNN-RNN-based approach to recognize the eight classes of chest illness and attained an AUC score of 0.753, whereas the method in [55] utilized the boosted cascaded convents algorithm with an AUC result of 0.778. Moreover, the methods in [56] and [57] acquired AUC values of 0.801 and 0.821. Chen et al. [59] selected a residual approach to recognize the various types of chest diseases, with an AUC score of 0.838. Comparatively, the CXray-EffDet model attained the highest AUC score of 0.908. 

Moreover, we further assessed the performance of the proposed method in the forms of the IOU metric with the new methods given in [58,59,60], and the obtained scores are depicted in Table 8. It is clear from the performance evaluation given in Table 8 that our work obtained the highest IOU score of 0.834.

The work discussed in [58] utilized a CNN approach to classify eight types of chest abnormalities with an IOU score of 0.569, and the works in [59] and [60] also utilized DL frameworks, with IOU scores of 0.746, and 0.728, respectively. Descriptively, the compared methods attained an average IOU score of 0.681, which was 0.834 for our work. Hence, the CXray-EffDet model provided a performance gain of 15.27% for the IOU metric.

It is fairly evident from the assessment that the suggested scheme for the diagnosis of chest illnesses is superior to more recent methods in terms of both the IOU and AUC measurement metrics. The higher meaningful feature-calculation capability of our approach, which helps it to effectively recognize all kinds of diseases, is the main cause of the presented solution’s strong recognition capacity. In contrast, the techniques in [54,55,56,57,58,59] have a structure that is highly complex, contributing to the problem of framework over-fitting. Additionally, the techniques are ineffective in precisely capturing visual features, because they cannot handle various distortions of suspected samples, such as fluctuations in light and color. In contrast, our approach is more successful at addressing the transformation variations in the X-ray medical image modality. As a conclusion, it can be said that the CXray-EffDet framework given here is more accurate in identifying and classifying chest diseases from X-ray images.

## 4. Discussion

Abnormalities found in the chests of humans impose a serious threat to their lives, as such diseases at the advanced level can cause the death of patients. Researchers are putting effort into introducing computer-aided solutions for the timely and reliable recognition of chest diseases from X-ray samples. However, very little attention has been paid to introducing such approaches that can not only determine the class of an abnormal chest area, but also exactly locate the diseased region, which can assist doctors in better examining it and starting curative procedures. The accurate and effective identification and classification of particular abnormalities in the chest using the X-ray images is a complicated problem because of the complex structural properties of these samples, e.g., their large exposure dynamic range. Moreover, the presence of different image artifacts and huge inter- and intra-class similarities further enhances the complexity of chest disease detection and classification procedures. The aim of this study was to resolve these existing problems. We have proposed a DL approach called the CXray-EffDet model. Descriptively, we utilized the EfficientNet-B0-based EfficientDet-D0 framework to extract a nominative set of sample features and perform the recognition and classification task by categorizing eight types of chest diseases via the use of the X-ray images. A complex test of the model was performed on the NIH CXR database. Our approach gained a mAP score of 0.926, along with precision and recall rates of 90% and 92.36%. Both the qualitative and quantitative results show that our approach is proficient at both locating and classifying the numerous categories of chest diseases. Additionally, the approach is robust at detecting various image distortions and notable inter- and intra-category similarities.

## 5. Conclusions

In the presented approach, a DL framework called the CXray-EffDet model was proposed to recognize and classify eight categories of chest abnormalities from X-ray samples. More clearly, we used the EfficientNet-B0-based EfficientDet-D0 model to compute a distinctive set of sample keypoints and accomplish the classification task. Furthermore, the proposed architecture is economically resilient at categorizing a variety of CXR anomalies, since it uses a one-stage object identifier to recognize numerous chest diseases. To demonstrate the efficiency of the suggested strategy, we performed extensive investigations on the NIH CXR database. We acquired a mAP score of 0.926, along with precision and recall rates of 90% and 92.36%. The findings show that the presented model surpasses existing frameworks with respect to both computing cost and classification results. Additionally, the method can effectively classify the different kinds of chest illnesses under the occurrence of various image distortions and notable inter- and intra-category similarities. The proposed work performs well for chest X-ray disease classification. However, a little performance degradation was observed for images with a blur effect. In the future, we plan to cover this limitation. Moreover, we plan to evaluate this model for all fourteen categories of chest abnormalities and accomplish an evaluation of other DL architectures as well.

## Figures and Tables

**Figure 1 diagnostics-13-00248-f001:**
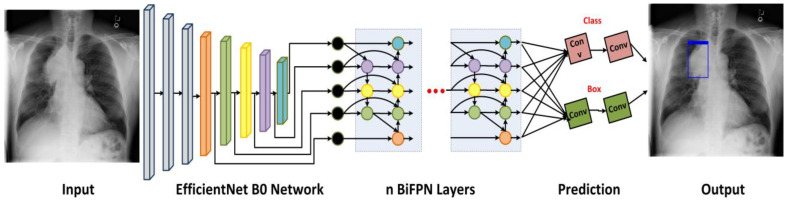
Pictorial description showing the basic flow of the proposed work.

**Figure 2 diagnostics-13-00248-f002:**
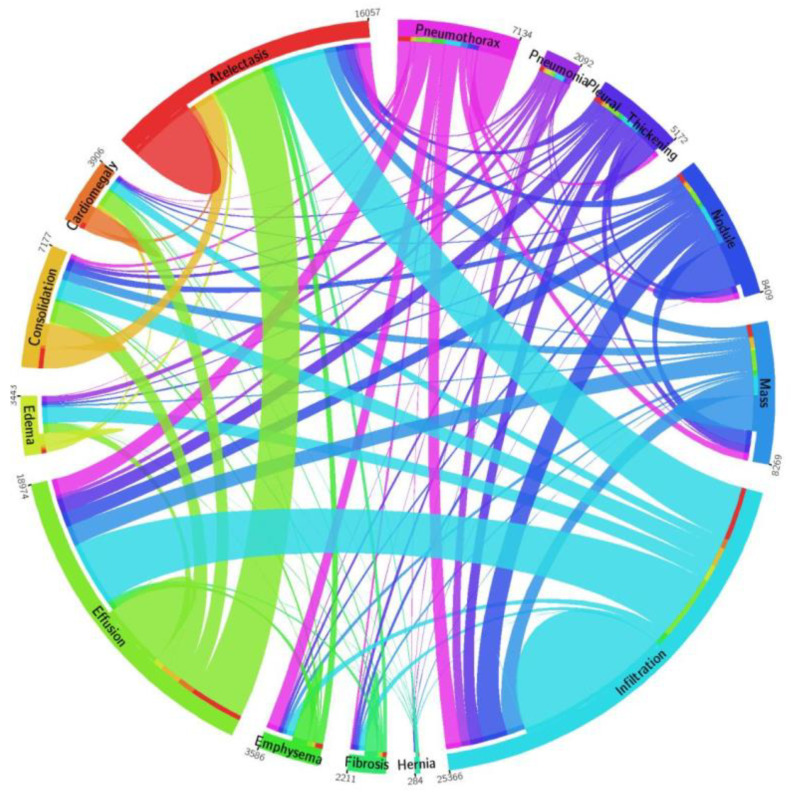
A visual description of the distribution of images from the NIH Chest X-ray repository [38].

**Figure 3 diagnostics-13-00248-f003:**
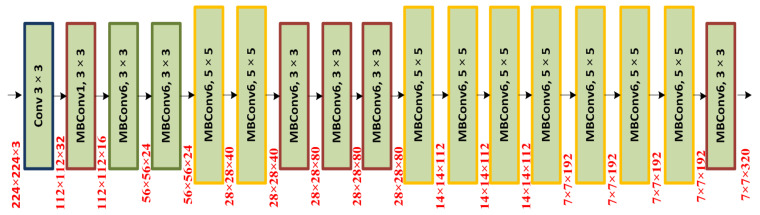
Architectural illustration of the EfficientNet-B0.

**Figure 4 diagnostics-13-00248-f004:**
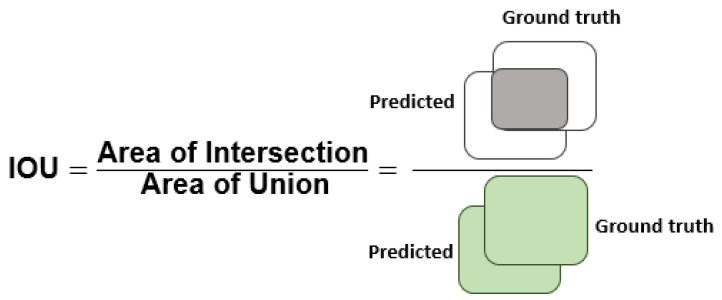
Pictorial description of IOU measure.

**Figure 5 diagnostics-13-00248-f005:**
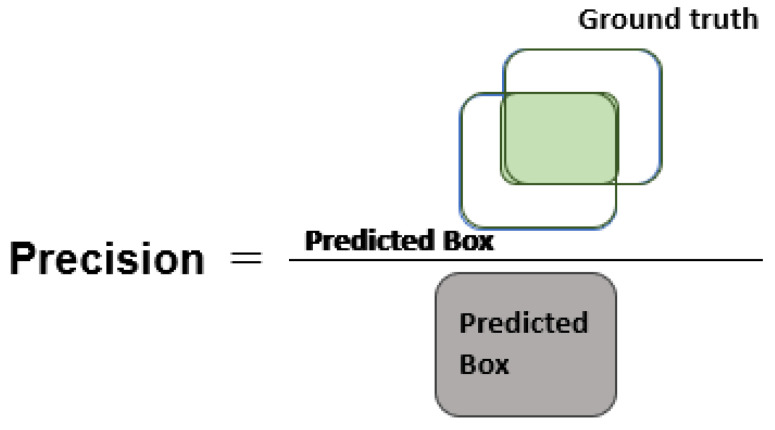
Pictorial description of precision measure.

**Figure 6 diagnostics-13-00248-f006:**
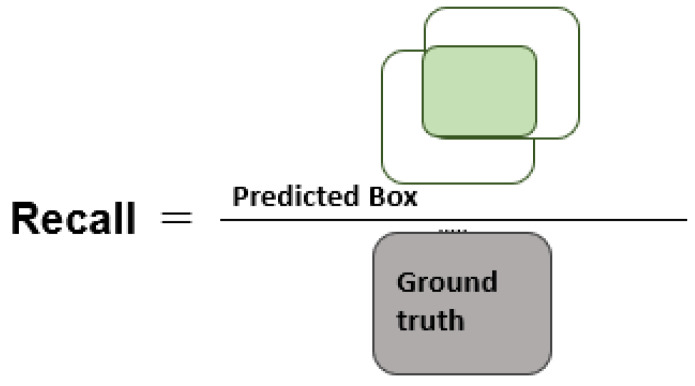
Visual description of recall metric.

**Figure 7 diagnostics-13-00248-f007:**
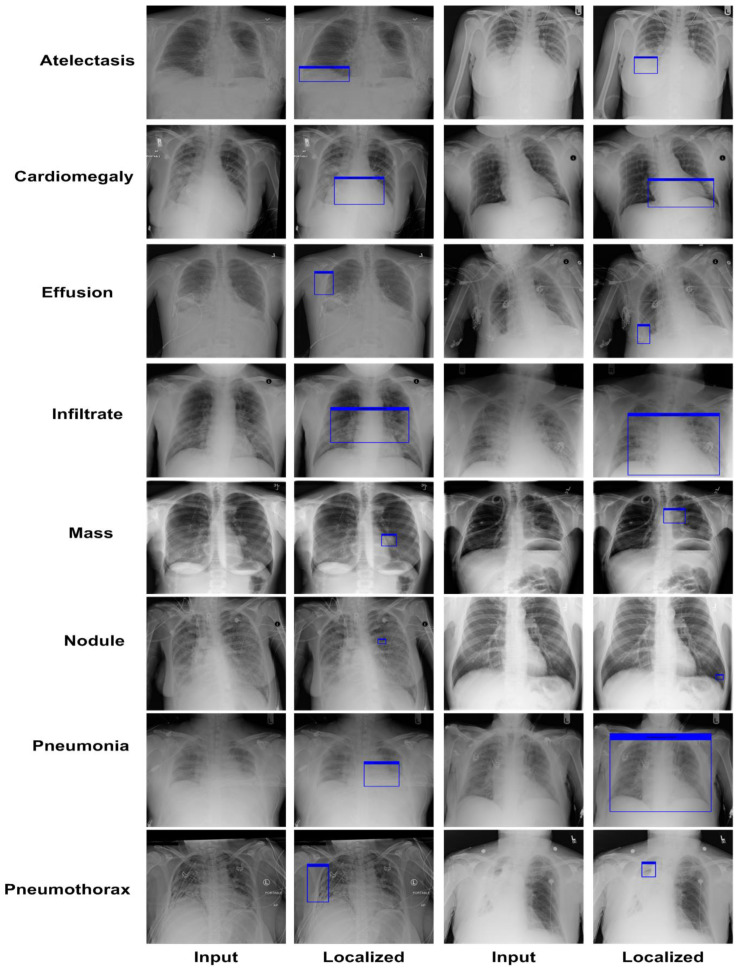
Localization results.

**Figure 8 diagnostics-13-00248-f008:**
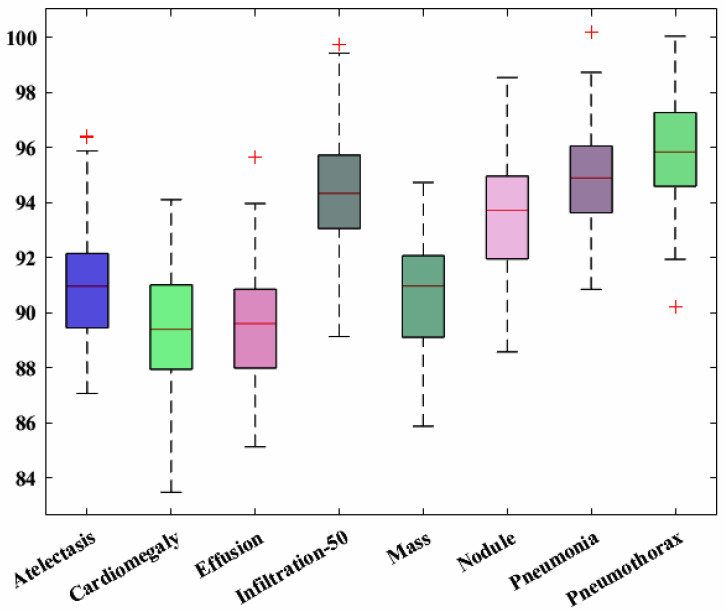
Category-wise precision score for the CXray-EffDet model.

**Figure 9 diagnostics-13-00248-f009:**
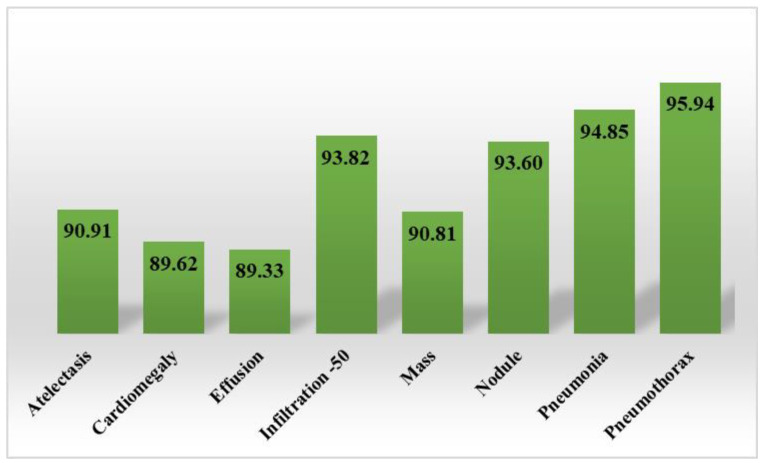
Category-wise recall scores for the CXray-EffDet model.

**Figure 10 diagnostics-13-00248-f010:**
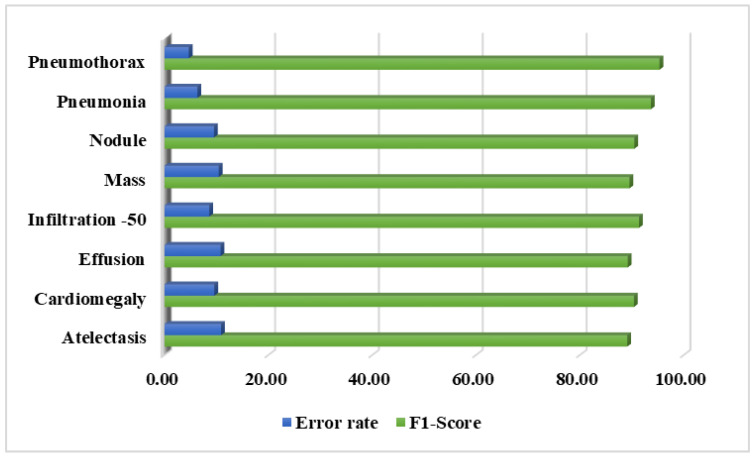
Category-wise F1-measure along with attained error rates for the CXray-EffDet model.

**Figure 11 diagnostics-13-00248-f011:**
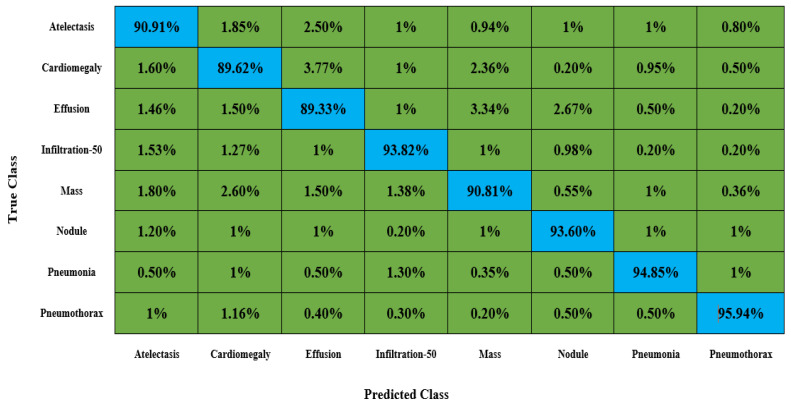
Attained confusion matrix results for the CXray-EffDet model.

**Figure 12 diagnostics-13-00248-f012:**
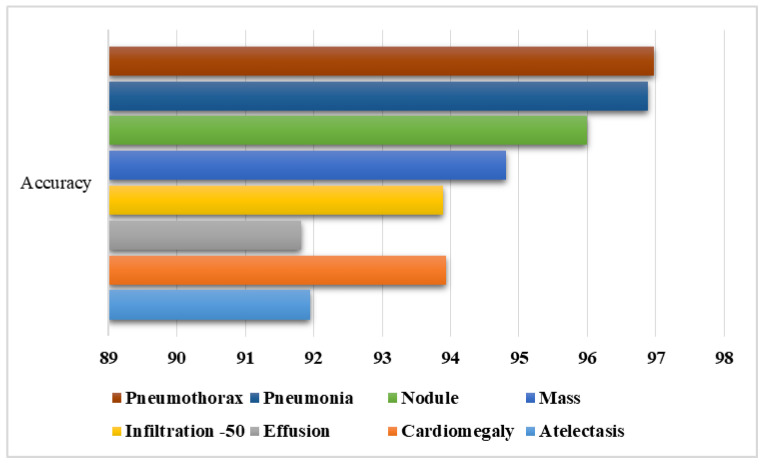
Category-wise accuracy results for the CXray-EffDet model.

**Table 1 diagnostics-13-00248-t001:** An evaluation of the existing work.

Reference	Model	Performance	Limitations Identified
[3]	VGG-16 XceptionNet	Accuracy = 87% (VGG16) Accuracy = 82% (XceptionNet)	Performance needs improvements.
[17]	AlexNet	Accuracy = 97.27%	Performance degrades for unseen cases.
[19]	GoogleNet	Accuracy = 80%	The classification results can be enhanced by including the segmentation technique to permit the model to extract more illness-related characteristics.
[20]	Deep Net	AUC = 80.10	Classification results require improvements.
[21]	Deep Net	Accuracy = 87%	Performance degrades for distorted images.
[22]	Ensemble method	Accuracy = 8462%	Classification results require improvements.
[23]	Deep Net	Accuracy = 92.4%	Performance degrades for distorted images.
[24]	DenseNet andmulti-scale attention network	AUC = 85	The performance can be improved further.
		AUC = 81.50	
[25]	DenseNet-121 DenseNet-169	AUC = 81.70	The framework is economically inefficient.
		AUC= 77.50	
[26]	ResNet-152	AUC= 78.10	The framework is economically inefficient with high inference time.
[27]	ResNet and visual AM	AUC = 81.90	The technique relies on the accessibility of box annotations.
		AUC = 91.66	
[28]	DenseNet MobileNetV2	AUC = 92.40	Performance degrades for unseen cases.
		AUC = 90	
[29]	ResNet-50	AUC = 96.90	The model is unable to tackle noisy images.
[30]	Deep Net	Accuracy = 94%	Requires evaluation on a complex dataset.
[31]	V2-EfficientNet	Accuracy = 82.15%	Performance degrades with an increase in disease classes.
[32]	ResNet-38	AUC = 0.822	Classification results need enhancement.
[33]	Modified VGG-19	Accuracy= 98.05%	Requires evaluation on a complex dataset.
[34]	ResNet and DenseNet	AUC =0.8398 (ResNet) and 0.8392 (DenseNet)	The model is unable to tackle distorted images.

**Table 2 diagnostics-13-00248-t002:** Details of the used hyper-parameters.

Parameters	Value
Total epochs	25
Learning rate	0.001
Batch size	16
Value of confidence score	0.5
Value of unmatched score	0.5

An in-depth explanation of all three units is given below.

**Table 3 diagnostics-13-00248-t003:** Comparison of the presented work with base techniques for the AUC measure.

Framework	**Atelectasis**	**Cardiomegaly**	**Effusion**	**Infiltration**	**Mass**	**Nodule**	**Pneumonia**	**Pneumothorax**	Average
**AlexNet**	64	69	66	60	**56**	65	55	74	64
**GoogLeNet**	63	70	69	61	54	56	59	78	64
**VGG16**	63	71	65	59	51	65	51	63	61
**ResNet50**	71	81	74	61	**56**	72	63	79	69
**Proposed**	91.30	97.10	94.60	79.60	92.50	85.70	88.10	97.50	90.80

**Table 4 diagnostics-13-00248-t004:** Comparison of the presented approach with base techniques for the entire database.

Model	Precision (%)	Recall (%)	Accuracy (%)	F1-Score (%)
**AlexNet**	65	66.14	67.45	65.57
**GoogLeNet**	69.53	71.88	70.35	70.69
**VGG16**	72	74.32	75.41	73.14
**ResNet-50**	77	75	77.63	75.99
**Proposed**	90	92.36	94.53	91.16

**Table 5 diagnostics-13-00248-t005:** Comparative assessment of the presented model with object-recognition frameworks.

Model	mAP	Test Time (sec/img)
Fast RCNN	0.65	0.28
Faster RCNN	0.77	0.25
Mask RCNN	0.79	0.23
RetinaNet	0.63	0.27
CenterNet	0.82	0.25
**Proposed**	0.926	0.20

**Table 6 diagnostics-13-00248-t006:** Comparative assessment of the presented model with ML-based algorithms.

Classifier	AUC
Deep-Keypoints + SVM [53]	74.50
Deep-Keypoints + KNN [53]	72.10
**Proposed**	**90.80**

**Table 7 diagnostics-13-00248-t007:** Comparative assessment of the presented model with new techniques for the AUC metric.

Approach	**Atelectasis**	**Cardiomegaly**	**Effusion**	**Infiltration**	**Mass**	**Nodule**	**Pneumonia**	**Pneumothorax**	Average
[54]	0.73	0.84	0.79	0.67	0.73	0.69	0.72	0.85	0.753
[55]	0.76	0.91	0.86	0.69	0.75	0.67	0.72	0.86	0.778
[56]	0.79	0.87	0.88	0.69	0.81	0.73	0.75	0.89	0.801
[57]	0.81	0.92	0.87	0.72	0.83	0.78	0.76	0.88	0.821
[59]	0.84	0.93	0.88	0.72	0.87	0.79	0.77	0.90	0.838
**Proposed**	0.94	0.95	0.93	0.92	0.93	0.84	0.84	0.91	0.908

**Table 8 diagnostics-13-00248-t008:** Comparative assessment of the presented model with new techniques for the IOU metric.

Approach	**Atelectasis**	**Cardiomegaly**	**Effusion**	**Infiltration**	**Mass**	**Nodule**	**Pneumonia**	**Pneumothorax**	Average
[58]	0.69	0.94	0.66	0.71	0.40	0.14	0.63	0.38	0.569
[59]	0.72	0.96	0.88	0.93	0.74	0.45	0.65	0.64	0.746
[60]	0.71	0.98	0.87	0.92	0.71	0.40	0.60	0.63	0.728
**Proposed**	0.81	0.99	0.94	0.96	0.81	0.63	0.78	0.75	0.834

## Data Availability

The dataset used in this work is publicly available.

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
