# Peer review of "CXray-EffDet: Chest Disease Detection and Classification from X-ray Images Using the EfficientDet Model"

_diagnostics, 2023, doi:10.3390/diagnostics13020248_

Round 1

Reviewer 1 Report

There are few  improvement require in manuscript in context of grammatical and technical perspective. Therefore, these issues must me properly address in next version.

1.     The Abbreviation in last of manuscript must be updated in  second version of manuscript.

what kind of complicated issues of Chest X ray highlighted by authors ?? for reference statement in abstract: "

 However, reliable detection and classi-22 fication of particular illnesses in chest X-ray samples is still a complicated issue because of the 23 complex structure of radiographs. "

In introduction section,  contribution 4 reflects the proposed approcah is CXray-EffDet is the computational effective , what's that mean?? if it related to computational cost then authors clear justify in your experimental results in terms of time complexity and space complexity . 

Authors must incorporated  the BIG o notation in experimental work of proposed approach for refection the computational effective 

 The novelty in proposed approach is limited , authors needs to justify for the publication.

Why the need of proposed approach?

What are the limitation in proposed approach?

Future work missing in the proposed approach

Dataset description clearly missing in this manuscript? either it's public or exclusive ?

  Format of the references must be relook, there are few ambiguities in few references

The contributions of each author must be clearly highlight

Author Response

Response sheet attached. thanks

Reviewer 2 Report

Include a list of four to ten key words after the Abstract
The aim is not clear in the abstract section
Mention major contribution and paper organization clearly
More recent literature survey required, and compare with them
Improve the results and discussion section, add more sentences for proper justifying the works
Try to improve the usage of English grammar. The formatting, grammar and typo errors should be carefully checked before processing this article.
The introduction of the article is very concise and short. From the Introduction section 1, it is rather unclear what gaps the manuscript attempts to explore and the overall added value of the paper. The contribution of the article should be highlighted clearly in the introduction section along with organization of paper.
The logical structure of this paper is good. Most importantly, the innovation of this paper is novel, and also, I feel that it meets the requirements of this journal.
Consider relevant works 10.1016/j.wndm.2014.09.002, https://doi.org/10.1007/s11042-022-13499-3

Add future scope of the works

Author Response

Response sheet attached. thanks

Round 2

Reviewer 2 Report

Authors addressed well

Author Response

Thanks for accepting our manuscript.